# Impact of Trade, FDI, and Urbanization on Female Employment System in SAARC: GMM and Quantile Regression Approach

Elvira Nica [1,*], Milos Poliak [2], Cristina Alpopi [1], Tomas Kliestik [3], Cristina Manole [1] and Sorin Burlacu [1]

1   Department of Administration and Public Management, Faculty of Administration and Public Management, Bucharest University of Economic Studies, Piața Romană, 010371 Bucharest, Romania
2   Department of Transport Technology and Logistics, Faculty of Operation and Economics of Transport and Communications, University of Zilina, 01026 Zilina, Slovakia
3   Department of Economics, Faculty of Operation and Economics of Transport and Communications, University of Zilina, 01026 Zilina, Slovakia
*   Correspondence: elvira.nica@ase.ro

**Abstract:** The fundamental objective of this research is to learn how trade liberalization, male employment, urbanization, and foreign direct investment (FDI) affect women's participation in the labor force. To continue, this study aims to determine the effects of trade and other factors on women's employment in three distinct sectors (i.e., agriculture, industry, and service). From 1991 to 2021, we analyzed data from eight SAARC countries. The study's theoretical foundation was the Cobb–Douglas production function. To better understand the connections between trade liberalization and the SAARC labor market, this paper used panel quantile regression (QR) and generalized method of moments (GMM) to empirically explore the key determinants of female employment in total and three sub-sectors. The QR method was used in the study because it looks at how variables affect each other beyond the data mean. Additionally, our data set does not follow a normal distribution, and the connection between the explained and explanatory factors is non-linear. Trade openness has a beneficial effect on total female employment throughout system GMM and all quartiles. Total female employment also benefits from an increase in GDP and FDI. However, women's access to the workforce is hampered by urbanization. Many strategies for increasing women's participation in the workforce across three sectors are addressed in this article. The major finding of this study is the rate of change in female employment across three industries. Women's participation in the service and manufacturing sectors increases, whereas their participation in agriculture decreases, as a result of increased trade openness. Although these studies can assist policymakers in choosing the best feasible trade adjustments, they will also add to diverse academic and policy discussions on trade liberalization and its gender consequences. Since trade has become more accessible, more and more women are entering the workforce. Therefore, workers should acquire industrial and service-sector-related competencies.

**Keywords:** employment systems; female; international trade; FDI; SAARC; quantile regression; urbanization



## 1. Introduction

The rapid growth of Asia's exports is a key factor in the region's recent economic success [1]. With the quickening pace of globalization, the emerging nations in Asia have adopted a wide variety of trade openness policies to lower their trade barriers and costs. Asia's affiliation with the World Trade Organization (WTO) has improved the region's trade openness. The total value of all merchandise exchanged worldwide increased from USD 0.8 trillion in 1980 to USD 30 trillion in 2013. It is also essential to remember that exports have increased more quickly than imports in many Asian nations [2]. Although south Asian nations have achieved tremendous economic progress, they are under enormous pressure to offer more and better employment to meet their rapidly expanding populations

and maintain their benefits. Low labor force participation rates, a persisting gender gap in employment, and a greater chance of unpaid work are all signs that women are being excluded from the labor market. For south Asia's economy to flourish, more women must be employed in productive roles, and those barriers to their participation in the labor force must be removed [3].

Women's economic participation has many positive effects. Positive development spillover effects of women's employment contribute to economic advantages, such as more political participation and a more robust "voice" for working women in the home. For instance, research in Bangladesh has found that women who work in the garment industry have better health outcomes, are put off having children, and marry later in life [4]. It has been shown that families in India with a female earner working in the business processing sector are more likely to prioritize their daughters' health and education [5]. Female employees are in high demand in Mexico's industrial sector due to the country's second-highest dependence on female workers, only behind south Asia. This is mostly due to the country's increased emphasis on promoting women's and children's health. Despite SAARC's economic progress, women are still under-represented in political and public life. Even though SAARC is one of the world's fastest emerging areas, only 23.6% of workers are female (men make up 80% of the workforce) [6]. There are several barriers preventing women from fully engaging in the workforce, including, but not limited to, gender-based violence, low levels of education for females, and gender rules that confine women's mobility. Women's labor force engagement in the SAARC area is on the rise. Some SAARC countries began lowering tariffs in the 1980s in response to GATT, WTO, and WB discussions. The 1990s saw profound changes in this area due to globalization and free trade. They pushed for an export-driven economy and designated special zones in the 1990s (EPZs). It was generally accepted that these reforms would significantly impact the growth of the labor market, the diminishment of poverty, increases in wages, the expansion of the economy, exports, and other outcomes [7,8]. Newcomers can take their cue from the rapid development seen in Hyderabad, Andhra Pradesh, India. In less than two decades, Andhra Pradesh went from having a sluggish economy based largely on agriculture to being a major service center. Once a straggler in its industry, it is now at the forefront. Between 1998 and 2008, the IT sector in Hyderabad grew eight times and added 200 percent more jobs as a consequence of a 45-fold growth in service exports. The growth of service-based economies is not unique to the southern parts of India and the SAARC countries. The average yearly growth rate for SAARC from 2000 to 2007 was just under 7%, making it competitive with east Asia. Trade policy changes have boosted the SAARC economy's size and increased the number of people actively seeking employment in the region [9]. This work explores the influence of trade liberalization on three aspects of women's labor market outcomes (including services, industries, and agriculture). The study aims to learn more about two important economic factors: international commerce and women's labor force engagement. About 32% of the world's impoverished, or 300 million, inhabit SAARC. Women in the area face obstacles in accessing the education, training, and employment opportunities they need to thrive. Additionally, acts of aggression against women and the marriage of minors are common [10].

Men hold approximately three times as many full-time jobs as women in the SAARC region (only 28.3% of women are in full-time employment). There are fewer opportunities for women to enter these industries and advance their careers. Despite significant economic growth, FLF rates in countries such as India have risen from 34.1% in 1999–2000 to 286% in 2016 [11]. There is a wider gap between the sexes in SAARC than in any other region by 18 percentage points regarding account ownership at official financial institutions. In India, small, medium, and micro-sized entities operated by women account for about USD 158 billion. Female cell phone ownership in SAARC is 38% lower than male ownership, the greatest digital divide in the world. Trade liberalization promotes the free movement of products, services, and capital, allowing nations to focus on areas where they have a competitive advantage. A government might save money and resources by focusing on

exporting commodities rather than importing them. Exporting can lead to lower average costs due to increased specialization and economies of scale. More productive use of resources, the development of economies of scale, and gains in productivity all contribute to rising real incomes.

There are two ways in which women would benefit from more trade openness. As a result, governments will have more money, and women will have more money in their everyday lives. Increases in GDP and per capita income from enlarging a country's commercial borders could fund higher levels of investment in areas such as public education and healthcare. A country's ability to meet the requirements of its citizens, such as social security, education, and healthcare, may be impacted by an increase in commerce on a macroeconomic scale. Microeconomic studies have shown that more trade openness leads to increased incomes, average income, overtime opportunities, and individual job options. Free trade expansion helps boost industrialization, capital investment, and output. It leads to better health for females. When women's wages, job opportunities, educational attainment, and job stability all rise, it becomes less attractive for them to quit the workforce to take care of household chores, have children, or pursue other non-work pursuits. App and Rees [12] and Engelhardt and Prskawetz [13] are examples.

The research carefully selects variables, which are highly correlated with women's employment. Increases in export capacities and trade liberalization policies have contributed to a dramatic expansion of trade [14]. As a result, women's ability to work has been increasing. Recently, export processing zones (EPZs) have been booming in the SAARC area. On the other hand, the female wage rate is lower than that of males in informal sectors of the south Asian region. Similarly, FDI is flourishing in south Asian countries. Women's employment opportunities are directly related to how much FDI enters a country. In urban areas, women are more secure working than in rural areas. Therefore, urbanization is essential when we consider women's employment. On the other hand, the more per capita income rises, the more family expenses also rise. The rising family expenses push women to access the job market. In contrast, male participation in the labor force encourages women to participate in the job market. In addition, the research selects eight SAARC countries because trade is booming in SAARC. Additionally, in SAARC countries, women's employment is rising. Urbanization and per capita income are sharply rising in south Asia. To the best of our knowledge, such work has not been conducted in SAARC.

The contributions of this research are multiple. First, the aim of this work was to determine the influence of trade, FDI, urbanization, GDP, and the male labor force on female employment in SAARC. Secondly, the research applied the Cobb–Douglas production function in the theoretical setting. The Cobb–Douglas production function is very helpful when we want to investigate an input–output or a cause–effect relationship. Third, the research also applied the cross-sectional dependency (CSD) test. Previously, most studies ignored the CSD problem. SAARC countries are highly interlinked with each other by trade, tourism, religion, politics, culture, and other factors. Therefore, there is a probability that the CSD problem is present here. Fourth, the research applied GMM and quantile regression to establish the interaction between dependent and independent variables. GMM helps in removing endogeneity and autocorrelation. On the other hand, QR helps when data are skewed and non-normally distributed [15,16]. Fifth, another major purpose was to investigate women's participation in three sectors: service, industry, and agriculture. In view of the booming trade, it would be useful to determine which sectors attract more women and how women switch from one sector to another as a result of trade.

The rest of the study is set out as follows. An analysis of the relevant literature is provided in Section 2. The facts, theoretical foundation, and econometric analysis are dissected and explored in depth in Section 3. The results and findings are presented and analyzed in detail in Section 4. Section 5 details the discussion portion of the paper. The conclusion is summed up in Section 6. In Section 7, we talk about policy implication. The last portion, or Section 8, shows the limitation and future research of this paper. The list of

south Asian nations included in this study can be found in Appendix A. Furthermore, the collection of acronyms can be found in Appendix B.

## 2. Literature Review

The literature on international economics and growth has devoted significant attention to women's issues in the workplace and international trade [17]. If the FLFPR grows, this signifies a rise in the available labor and, consequently, an increase in the nation's production capacity, which is favorable for the forthcoming output and growth of the economy [18]. If south Asia lowered trade barriers, more jobs might be created, especially in export-focused businesses. They favor low-cost female laborers, but increased competition from freer commerce could hurt their competitiveness due to lower educational levels. It is widely understood that the expansion of commerce results in winners and losers due to globalization's reallocation of resources. Gender disparities in pay and the job market are just two indicators of women's performance in male-dominated fields. Asian countries share a common cultural heritage, including a tradition of conservatism in societal attitudes regarding women in the workplace [19]. Furthermore, export reliance has a specific threshold impact on female employment [20], which indicates that exports would increase women's labor to a certain level, which is consistent with the discriminatory argument. Thus, a healthy dose of exports would serve to boost female employment. Pakistan's workforce now has a higher percentage of women because of increased trade openness, as Hyder and Behrman [21] explain in greater detail. There has been a no-figure rise in the number of women working in Bangladesh's and Sri Lanka's export-focused garment sectors. At first impression, trade liberalization seems to promote women's involvement in the beginning stages of export-driven activities in southeast and east Asian nations. This trend may revert in the later stages. This is because of structural adjustments and technical maturation of export products, referred to as "de-feminization" [22]. Even though imports and exports each account for a unique percentage of total trade, this difference is rarely accounted for in the existing literature. In general, the nexus between female employment and trade in developing Asian nations is overlooked in academic publications. The alteration of the external structure and time-varying properties of time series are also disregarded when making predictions due to the linearity assumption. The consequences would be a less rigorous analysis [23].

According to research by Chatterjee et al. [24], which is consistent with this story, rural women's involvement in the labor force rate in India fell by 12 to 14 percentage points between 2004–05 and 2010–11, probably as a result of a significant reduction in farming jobs. The "U-shaped" association between the education of women and labor force inclusion in India is discussed by Chatterjee et al. [25]. Demand-side explanations are offered for the disparate employment prospects of women with varying levels of education. They argue that discrimination against women in the workplace, namely in traditionally female-dominated fields, such as office work and retail sales, is the root of the problem. Despite a doubling of college graduates in the labor force, Klasen and Pieters [26] argue that the decline in educated women's labor force involvement in urban India is due to their absence of representation in fields, which are well suited to their education and training, such as the professional and business services. Evidence suggests that women will seize new employment opportunities when economies shift away from agriculture. Due to the expansion of Bangladesh's export-oriented textile business, the number of women working outside Dhaka's urban core has increased [4].

The large growth in women's formal work in Bangladesh can be traced back to their exposure to international trade immediately before and after its emergence [27]. Increased female employment may result from women's education and skill development investments. According to Mehrotra and Parida [28], women's lower education levels make them less competitive for positions in the manufacturing sector, even though this is one of the fastest growing industries in need of trained workers. According to Juhn et al. [29], improved export options for Mexican manufacturers helped reduce the gender gap in blue-

collar manufacturing jobs. Topalova [30] looked at how tariff reductions in one industry might have a rippling impact on a region's economy based on changes in the industrial mix before liberalization. After considering the pattern changes and time-invariant regional unobservable characteristics, it is possible to compare the development of trade protection across micro-regions to assess the effect of trade openness on female and male labor market output.

Besamusca et al. [31] were the first to compile data at the national level on women's labor force involvement across age groups and individual factors. The ages of the participants influence the conclusions they reach. Women are more likely to return to work after having children if they are allowed paid leave. Female employment correlation with other factors can be investigated using several independent variables. Krishna et al. [32] used these data to pinpoint every factory in the Istanbul metropolitan area. Their calculations suggest a larger and more significant elasticity of labor demand for women only. Utilizing data from a survey of the Nepal labor market conducted in the late 1990s, Fofana et al. [33] found that women were more open to these changes than men and that their increased participation in market activities cut into their free time. From 1991 to 2009, Anyanwu [34] explored the influences of macroeconomic factors on youth unemployment in 48 African countries. His findings from FGLS indicated that FDI was linked to higher rates of youth unemployment in Africa, while openness was connected to lower rates of youth unemployment in the continent. According to Awad's [35] research, fifty African nations experienced significant increases in youth unemployment between 1994 and 2013 due to globalization. One of the variables he needed to manage was the employment situation. He used the Arellano–Bond (AB) estimator and generalized method of moments (GMM) technique to determine that easier access to international markets reduced youth unemployment. According to Selwaness and Zaki [36], the favorable effects of exports on employment in north Africa and the Middle East have been muted by the region's rigid labor markets.

Furthermore, due to trade liberalization, research has yet to be conducted on the issue of women's labor force participation in SAARC's agricultural, industrial, and service sectors. Very few research works are based on primary data with limited observations. In addition, no research work simultaneously examines the effects of FDI, urbanization, male employment, and trade openness on total female employment in the SAARC area. This study tries to reveal how trade openness affects several sectors of the economy, including agriculture, manufacturing, and the service sector. Similarly, the mechanism of female employment switching from one sector to another as a result of trade is discussed in this research article. The research also looks into the CSD problem. Previous researchers ignored the CSD problem. The member nations of SAARC have extensive ties to one another in the areas of commerce, tourism, religion, politics, culture, and other spheres. Therefore, it is likely that we are confronted with the CSD issue. Neglecting CSD will lead to misleading findings. Additionally, previous researchers entirely avoided the GMM and QR methodology. GMM is very important here because there is a high probability of the presence of causality between trade openness and women's employment. The quantile regression approach is advantageous because it helps us make sense of outcomes that are not normally distributed and have non-linear interactions with predictor factors by allowing us to comprehend the correlations between variables beyond the mean of the data. However, in this study, the research employed the system GMM and the QR approach to reveal valuable insights into women's employment and its influential elements, such as trade openness, urbanization, male employment, and FDI, in SAARC countries, which had not been performed previously by any other researchers.

## 3. Methodology

### 3.1. Data and Descriptive Evidence

This paper's variable lists are presented in Table 1. In addition to GDP per capita, urban population, male employment, and foreign direct investment (FDI), we use these variables to quantify the effects of trade openness on female employment. They are all

connected to women's labor force participation. The World Development Indicator (WDI) data utilized in this study were obtained from the World Bank [37].

**Table 1.** Description of variables.

| Variable Name | Log Form | Indicator Name |
|---|---|---|
| Female employment | lnFE | Rate of female labor force participation (percentage of the female population aged 15 and over) ILO (estimated ILO) |
| Females in agriculture | lnAGRI | % of women who work in agriculture (modeled ILO estimate) |
| Females in industry | lnINDUS | % of women who work in industry (modeled ILO estimate) |
| Females in service | lnSERV | % of women who work in service (modeled ILO estimate) |
| Trade openness | lnTO | Trade (% of GDP) |
| GDP per capita | lnGDP | GDP per capita (current USD) |
| Urban population | lnURBA | Urban population (% of the total population) |
| FDI | lnFDI | Foreign direct investment, net inflows (BoP, current USD) |
| Male employment | lnME | The participation rate of men in the labor force (percentage of men over 15) |

Source: WDI, World Bank (2022) [37].

Descriptive statistics for all variables are summarized in Table 2. The table summarizes the descriptive statistics for eight variables. The variables are: lnFE, lnTo, lnGDP, lnME, lnURBA, lnFDI, lnAGRI, lnINDUS, and lnSERV. The statistics include the number of observations (N), mean, standard deviation (sd), minimum (min), and maximum (max) values for each variable. The mean is the average value of the variable, while the standard deviation is a measure of how much the values of the variable differ from the mean. The minimum and maximum values give an indication of the range of values for each variable. The second row displays the log of the overall number of female employees. On the other hand, rows 3 through 5 display the log employment in each of the three industries.

**Table 2.** Synopsis of descriptive statistics.

| Variables | N | Mean | SD | Min | Max |
|---|---|---|---|---|---|
| lnFE | 224 | 3.525 | 0.531 | 2.526 | 4.416 |
| lnAGRI | 216 | 3.940 | 0.788 | 1.308 | 4.494 |
| lnINDUS | 216 | 2.435 | 0.668 | 0.405 | 3.759 |
| lnSERV | 216 | 2.968 | 0.612 | 1.825 | 4.332 |
| lnTO | 198 | 3.976 | 0.548 | 2.843 | 5.215 |
| lnGDP | 198 | 6.719 | 0.925 | 4.766 | 9.198 |
| lnME | 216 | 4.399 | 0.0669 | 4.209 | 4.508 |
| lnURBA | 208 | 3.206 | 0.327 | 2.217 | 3.840 |
| lnFDI | 180 | 13.24 | 2.625 | 9.828 | 18.07 |

Source: Author's computation.

Figure 1 depicts the level of trade openness and female employment in SAARC countries. The figure depicts how trade openness and female labor force involvement evolve. The figure shows that trade and female employment are both growing.

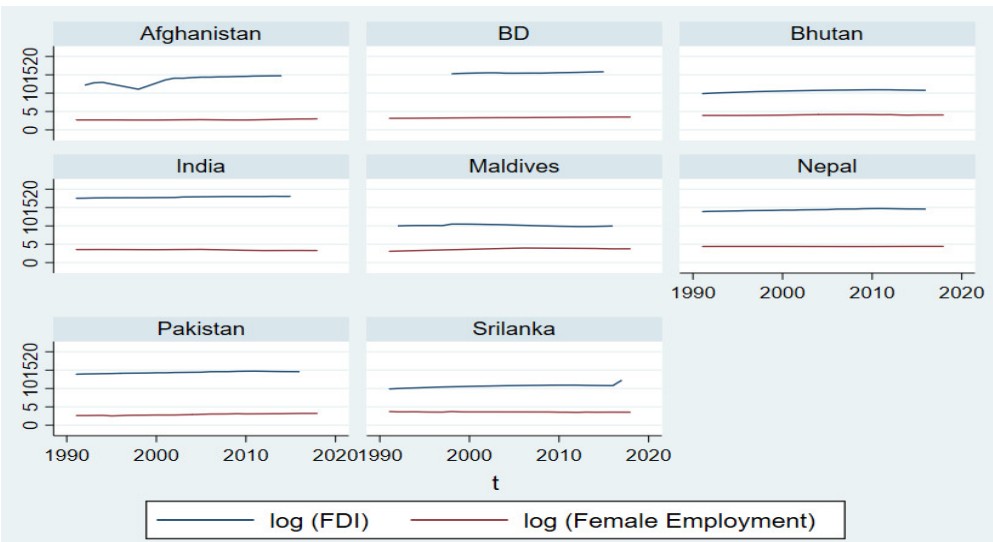

**Figure 1.** The progression of trade openness and female employment in SAARC countries. Source: Author's calculation.

Foreign direct investment (FDI) and female employment rates in SAARC nations are shown in Figure 2. This chart illustrates the rising tide of foreign direct investment and the number of women in the labor force.

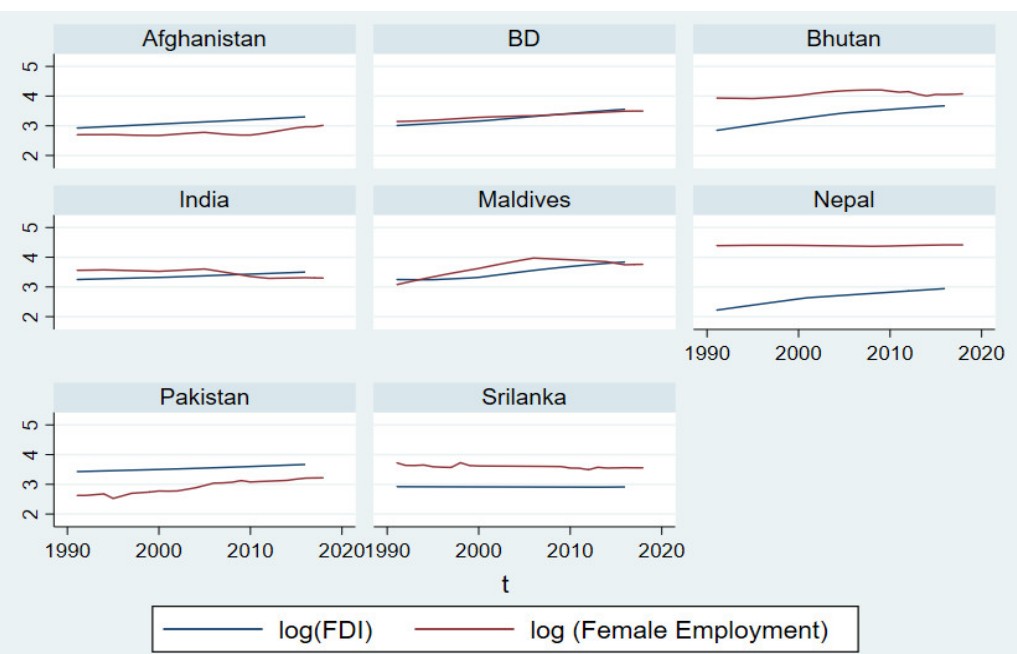

**Figure 2.** The progression of FDI and female employment in SAARC countries. Source: Author's calculation.

Figure 3 displays the relationship between urbanization and female employment in SAARC countries. The figure below shows how urbanization and female labor force participation have both evolved over time.

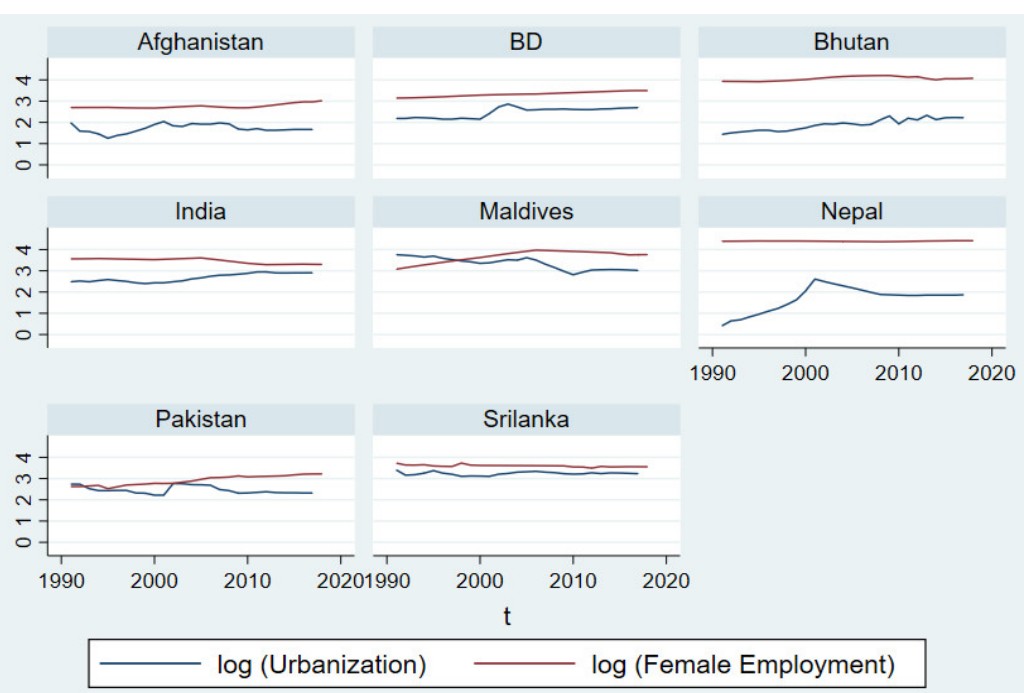

**Figure 3.** The progression of urbanization and female employment in SAARC countries. Source: Author's calculation.

*3.2. Theoretical Framework and Cobb–Douglas Model*

The lower production costs necessitate lower labor costs. Women's wages tend to be lower than men's in most SAARC countries. Historically, women from SAARC countries have had little or no opportunity to work in the service or industrial industries. Greater access to paid work in both formal and informal sectors was made possible because of increased trade opening. Increasing women's engagement in the labor force can be achieved through greater trade openness, which fosters economic growth, lowers the poverty rate, and improves gender equality by narrowing the wage difference. Our research aims to examine how trade openness impacts a variety of industries and to demonstrate the elasticity of different quantiles. This work aims to determine the effect of trade, income, male employment, urbanization, and FDI on female employment. This research consciously selected the independent variables because female employment highly depends on export and import, income level, urban area, foreign direct investment, and male participation in south Asia. Recently, export processing zones have boomed in Bangladesh, and most women are involved in the garment industry. Because of FDI, many multinational companies moved to Bangladesh and employed a lot of female labor. In urban areas, various job scopes are open for women. Likewise, working in an urban area is more secure than working in a rural area. Additionally, living in urban areas is expensive, making it challenging for a husband to maintain a family and a wife to try to support her husband economically.

Similarly, male participation encourages women to take up the jobs. Therefore, the variables we selected are highly related to women's employment. Women's participation is mainly found in the service, industry, and agriculture sectors. This study's secondary objective was to examine the impact of trade openness on women's involvement in these three industries.

Figure 4 shows that women's employment is associated with GDP, trade openness, FDI, urbanization, male employment, and other macroeconomic factors. In this context, the Cobb–Douglas production function is suitable for building a theory to meet the research purpose. The theoretical foundation of this investigation is the Cobb–Douglas production function. The following model is employed to understand the factors of female employment in SAARC countries from 1991 to 2019. The model describes the connection between

production output (women's employment) and production inputs (the many elements that contribute to women's employment) [38]. Quantitatively estimating the output (Y) as a function of total inputs of labor (*L*) and capital (*K*) is the goal of the Cobb–Douglas production function model. Simplifying the production function equation yields the following form:

$$\mathrm{Y} = f\left(L^a K^b\right) \tag{1}$$

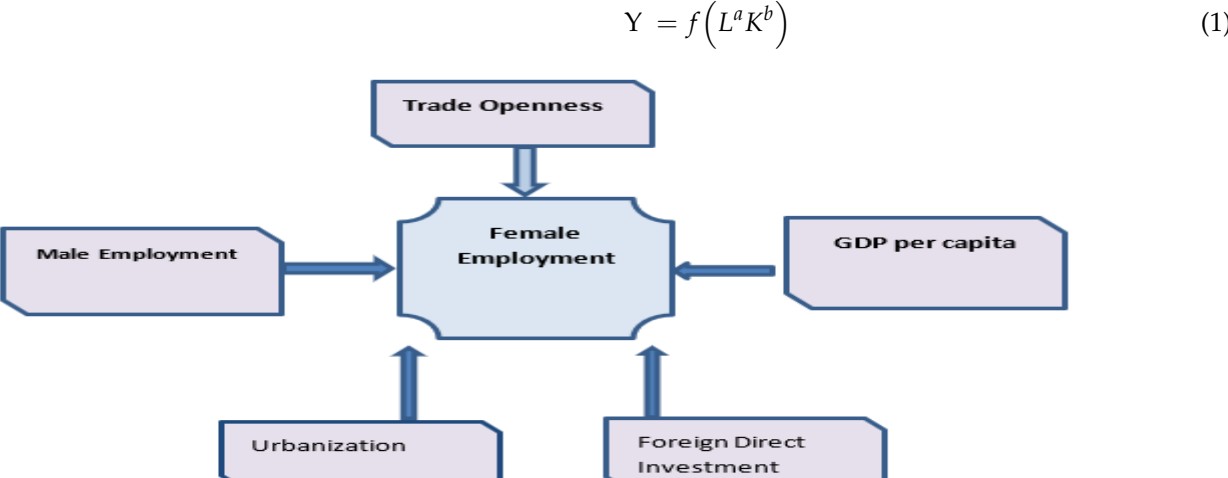

**Figure 4.** The framework of trade openness and other attributes. Source: Author's calculation.

In Equation (1), Y is the output, and *a* and *b* are the labor and capital output elasticities [38]. Where *a* + *b* = 1, the function is a homogeneous set with constant returns to scale [39]. The GMM and QR estimators are applied to estimate the parameters a and b. Female employment, or Y, is the dependent variable in this study, whereas GDP, FDI, trade, male employment, and urbanization are the independent variables (*L* and *K*), respectively. In this model, the input variables are developed by two sector variables (*L* and *K*). *L*, or labor-related variables, are urban populations, urbanization, and male labor force participation.

Similarly, *K*, or capital-related variables, are GDP, FDI, and trade. The Cobb–Douglas production function has been extended beyond its original confines of production, labor, and capital to include output versus input variables, influenced versus influencing elements, production versus participation factors, and supply versus demand factors [40–46]. In this paper, the Cobb–Douglas setting is well suited because female employment is the output or production factor; alternatively, trade, GDP, male employment, urbanization, and FDI are applied as the participating or influencing factors. Assuming a market-clearing scenario, wherein female employment is proportional to trade, GDP, male employment, urbanization, and FDI, Equations (2) and (3) provide a good match.

The baseline equation of our model is represented by the following reduced-form Equation (2).

$$\text{Female Employment} = f\left(Trade,\ GDP,\ Maleemployment,\ FDI,\ Urbanization\right) \tag{2}$$

Equation (3) is the extended form of the equations mentioned above:

$$\mathrm{FE}_{it} = \beta_0 + \beta_1 \mathrm{TO}_{it} + \beta_2 \mathrm{GDP}_{it} + \beta_3 \mathrm{ME}_{it} + \beta_4 \mathrm{URBA}_{it} + \beta_5 \mathrm{FDI}_{it} + \varepsilon_{i,t} \tag{3}$$

Equation (4) is more comprehensive by taking log values from both sides:

$$\ln \mathrm{FE}_{it} = \beta_0 + \beta_1 \ln \mathrm{TO}_{it} + \beta_2 \ln \mathrm{GDP}_{it} + \beta_3 \ln \mathrm{ME}_{it} + \beta_4 \ln \mathrm{URBA}_{it} + \beta_5 \ln \mathrm{FDI}_{it} + \varepsilon_{i,t} \tag{4}$$

$\mathrm{FE}_{i,t}$ is a measure of female employment in SAARC over time (t). TO is a measure of trade openness. The trade openness metric is the GDP export and import totals as a

percentage. ME denotes male employment. URBA and FDI indicate urbanization and foreign direct investment. The essential estimate parameters are $\beta_1$ to $\beta_5$ in this research.

In employment, labor force, and labor economics research, the Cobb–Douglas method was applied in various ways [47–51]. Our dependent and independent variables are well suited to the Cobb–Douglas model.

### 3.3. Econometric Framework

### 3.3.1. CSD Test

The research considers the cross-section dependency (CSD) test. SAARC countries are highly interrelated through trade, tourism, religion, and education. Therefore, the CSD test is essential here.

### 3.3.2. Generalized Method of Moments (GMM)

CSD is absent in the data. On the other hand, there may be endogeneity in the data because of trade openness, and women can enter the job market more. Inversely, because of women entering the job market, trade is increasing.

This study applied a dynamic panel data estimate model to analyze the correlation between independent factors and female employment. The predicted model is constructed by adding the dependent variable as a lagged addition to the regressors. This is the time of the delayed arrival, which the right-hand variables must consider. However, because of the interaction between the lagged dependent variable and the stochastic or constant individual-specific influence, the model's ordinal least squares (OLS) estimate will be inconsistent and skewed. Therefore, the fixed-effects model's within-transformation may be used to solve this issue. However, it is shown again that the static estimator is skewed due to the linkage between the lagged dependent variable and the disturbance component. There are many potential remedies to the dynamic panel model's bias issue, including Anderson and Hsiao's [52] introduction of the instrument variable (IV) technique. However, with heteroscedasticity present, the standard IV estimator could be more efficient. Hansen [53] advocated for the employment of the generalized method of moments (GMM) when individuals exhibit heteroscedasticity and autocorrelation. The methodology has succeeded in panels with low N and T (narrow temporal dimensions). The approach kicks out with an orthogonal deviation transformation or initial differences to mitigate the impact of outliers and account for the likelihood of non-stationary data [54]. The next step is to choose suitable IVs to solve the connection between the lagging response variable and the disturbance term. It has been suggested by Anderson and Hsiao [55] to use independent variables of the level form, which are at least twice as late as the IVs. After ensuring no serial correlation exists in the residuals, the optimal lag time is selected. According to Arellano and Bover [56] and Blundell and Bond [57], using the lagged level form of the explanatory variable as IVs can lead to issues, such as incorrect instrument choice, imprecision, and a country-specific impact, which is time-invariant in the first-difference equation.

For this reason, the GMM method was developed. The system GMM uses lagged levels and lagged first differences in level equations as instrumental variables. Under the null that these moment requirements are valid, the standard Hansen test produces an asymptotic distribution [53,58] and is used to ascertain whether or not the model is over-identified and whether the instruments utilized are valid. Hence, the system GMM technique is applied in this work to examine the effect of exogenous factors. Moreover, the instruments generated by system GMM have some limitations due to the small dimensions of the panel data set, particularly the small number of countries compared to the number of years [59]. Contrarily, many research works have applied system GMM when the number of countries was comparatively small to the number of years [60–64]. When there are fewer nations than years, the GMM system can be used as well [65–69]. In contrast, a significant number of studies implemented the GMM system in situations where the number of countries is

low relative to the number of years [60–64]. The current research formulates the function in the following manner using the system GMM method:

$$y_{it}=\alpha y_{i,t-1}+x'_{i,t}\beta+\gamma_i+\varepsilon_{it} \tag{5}$$

In Equation (5), the country is labeled $i$, and the time is labeled $y_{It}$ is the dependent variable (female labor force participation number). The exogenous variables are labeled $x'_{it}$. $\beta$ is the vector of estimation coefficients; $\gamma_i$ is the single effect; and $\varepsilon_{it}$ is the disturbance term.

### 3.3.3. Quantile Regression (QR)

To explore the influence of trade openness on women's involvement in the labor force, we use the QR strategy described by Koenker and Bassett [70] as the estimating methodology. The quantile estimator can be obtained by solving the following optimization problem. Quantile regression is used to describe the relationship between a group of predictors or independent variables and specific percentiles or quantiles of a target or dependent variable, which is most commonly the median. This method, compared to the ordinary least squares regression method, has two significant advantages. Quantile regression does not make any assumptions about the distribution of the target variable. Outlier observations have a lower likelihood of having an impact on quantile regression.

Buchinsky [71] points out that, in order to describe the feasible heterogeneous influences, we identify the qth-quantile ($0 < q < 1$) of the response variable as impermanent distribution, given a set of $X_i$ variables, as follows:

$$Q_q\left(\frac{y_t}{x_t}\right) = a_q + x_t \cdot \beta_q + u_t \cdot \alpha_q \tag{6}$$

In Equation (6), $y_t$ is the female employment rate over time, and $u_t$ denotes factors that the errors cannot observe. There is also an independent variable vector ($X_{it}$). According to Cameron and Trivedi [72], the following objective function must be used for Equation (7) inference based on qth quantile regression:

$$Q(\beta_q) = min_\beta \sum_{i=1}^{n} \| y_t - x_t\beta_q \| = min\left[ \sum_{y_t \geq x_i\beta} q|y_t - x_{it}\beta_q| + \sum_{y_t < x_{it}\beta} (1-q)|y_t - x_{it}\beta_q| \right] \tag{7}$$

Following Canay [73], the assessment process is divided into two sections. The restrictive mean of $u_t$ is determined and analyzed in the first stage. To continue, this element is removed from the primary response variable before the quantile regression evaluation is carried out.

## 4. Result and Findings

Table 3 shows the relationships between lnFE and lnTO, lnGDP, lnME, lnURBA, and lnFDI—0.2779, 0.0897, −0.1157, −0.4439, and 0.2988—all of which were significant at the 1% level. The findings showed that lnTO, lnGDP, lnME, lnURBA, lnFDI, and lnFE have substantial positive and negative relationships, implying that increases and decreases in the mentioned factors reduce and increase female employment. lnME and lnURBA showed a negative connection with lnFE, implying that female employment declines as the male labor force and urbanization grow. lnTO and lnGDP, lnME, lnURBA, and lnFDI, however, have correlations of 0.5053, −0.5123, 0.0494, and 0.3586. lnME and lnURBA have a negative correlation of −0.313. Because each variable is perfectly associated with itself, the correlation coefficients along the table's diagonal are equal to 1. These cells are useless in terms of interpretation.

**Table 3.** Variables' correlation matrix.

|  | lnFE | lnTO | lnGDP | lnME | LnURBA | lnFDI |
|---|---|---|---|---|---|---|
| **lnFE** | 1.0000 |  |  |  |  |  |
| **lnTO** | 0.2779 | 1.0000 |  |  |  |  |
| **lnGDP** | 0.0897 | 0.5053 | 1.00 |  |  |  |
| **lnME** | −0.1157 | −0.5123 | −0.642 | 1.000 |  |  |
| **lnURBA** | −0.4439 | 0.0494 | 0.539 | −0.313 | 1.000 |  |
| **lnFDI** | 0.2988 | 0.3586 | 0.481 | 0.656 | 0.090 | 1.00 |

Source: Author's computation.

Economists working with panel data must investigate CSD at the beginning. CSD helps to find out whether the variables are interdependent or not. The result will be biased if any panel research skips the CSD problem. The results of testing for CSD are shown in Table 4, where it is seen that there is no CSD problem in our data. Out of four tests, only one test is significant here. Therefore, we can apply the first-generation regression test here. Therefore, these countries have economic, social, and political similarities; lnFE, lnTO, lnGDP, lnME, lnURBA, and lnFDI all reveal that there is no CSD problem.

**Table 4.** CSD test.

| Tests | Test Statistics | *p*-Value |
|---|---|---|
| Pesaran CD test | 2.248 | 0.124 |
| Pesaran scaled LM | 0.42 | 0.570 |
| Friedman test | 1.214 *** | 0.254 |
| Breusch–Pagan LM test | 14.37 *** | 0.060 |

Note: Asterisk sign *** indicates a 1% level of significance. Source: Author's computation.

Three-unit root tests were carried out in the case of data in Table 5. The findings indicate that all variables are stationary at the first difference. As a result, we can use the quantile regression model we presented.

**Table 5.** Stationarity test.

|  | At Level |  |  | At First Difference |  |  |
|---|---|---|---|---|---|---|
| **Variables** | **Harris–Tzavalis** | **Im–Pesaran–Shin** | **Levin, Lin, and Chut** | **Harris–Tzavalis** | **Im–Pesaran–Shin** | **Levin, Lin, and Chut** |
| lnFE | 0.348 | 0.726 | −0.471 | −22.35 *** | −8.765 *** | −5.613 *** |
| lnTO | 1.842 | 2.594 | 3.70 | −21.44 *** | 9.13 *** | −7.29 *** |
| lnGDP | 0.748 | 1.145 | 0.362 | −19.10 *** | 8.956 *** | −5.15 *** |
| lnME | −1.256 | −0.863 | −0.073 | −27.19 *** | −9.33 *** | −7.88 *** |
| lnURBA | −1.198 | −0.736 | −0.559 | −21.93 *** | −9.177 *** | −7.82 *** |
| lnFDI | −1.205 | 0.617 | 0.545 | −31.52 *** | −9.769 *** | −7.72 *** |

Note: Asterisk sign *** indicates a 1% level of significance. Source: Author's computation.

Table 6 provides the paper's estimated effect of trade openness on female labor force participation. However, the effects of trade openness and other attributes on three subsectors are displayed in Tables 6–8.

**Table 6.** Impact of trade and other attributes on total female employment.

| Variables | S-GMM | Q25 | Q50 | Q75 |
|---|---|---|---|---|
| L.lnFE | 1.202 *** | | | |
| | (0.784) | | | |
| lnTO | 0.085 ** | 0.203 ** | 0.385 ** | 0.407 *** |
| | (0.170) | (0.158) | (0.170) | (0.125) |
| lnGDP | 0.232 ** | −0.0191 | 0.232 ** | 0.367 *** |
| | (0.104) | (0.0970) | (0.104) | (0.0767) |
| lnME | 0.345 | −6.198 *** | 0.345 | −2.700 ** |
| | (1.426) | (1.332) | (1.426) | (1.053) |
| lnURBA | −0.692 *** | −0.974 *** | −1.192 *** | −1.819 *** |
| | (0.234) | (0.219) | (0.234) | (0.173) |
| lnFDI | 0.0389 | 0.0620 | 0.0389 | 0.100 *** |
| | (0.0427) | (0.0399) | (0.0427) | (0.0316) |
| Constant | 2.338 | 33.68 *** | 2.338 | 16.22 *** |
| | (6.379) | (5.955) | (6.379) | (4.711) |
| Observations | 159 | 175 | 175 | 175 |

Standard errors enclosed by brackets. Asterisk sign **, *** for a *p*-value less than 0.1, 0.05, and 0.01. Source: Author's computation.

**Table 7.** Impact of trade and other attributes on female employment in the agriculture sector.

| Variables | S-GMM | Q25 | Q50 | Q75 |
|---|---|---|---|---|
| L.lnAGRI | 0.970 | | | |
| | (0.485) | | | |
| lnTO | −0.299 *** | −0.179 *** | −0.299 *** | −0.142 *** |
| | (0.122) | (0.145) | (0.122) | (0.0531) |
| lnGDPpc | −0.465 *** | −0.665 *** | −0.465 *** | −0.116 *** |
| | (0.0748) | (0.0885) | (0.0748) | (0.0325) |
| lnME | −3.360 *** | −5.627 *** | −3.360 *** | 0.552 |
| | (1.028) | (1.216) | (1.028) | (0.447) |
| lnURBA | −0.0347 | −0.206 | −0.0347 | −0.0426 |
| | (0.169) | (0.200) | (0.169) | (0.0734) |
| lnFDI | 0.0560 * | 0.117 *** | 0.0560 * | 0.0153 |
| | (0.0308) | (0.0364) | (0.0308) | (0.0134) |
| Constant | 21.45 *** | 32.86 *** | 21.45 *** | 2.673 |
| | (4.596) | (5.438) | (4.596) | (1.997) |
| Observations | 158 | 173 | 173 | 173 |

Standard errors enclosed by brackets. *p*-value less than 0.1, 0.05, and 0.01 indicated by asterisk signs *, ***. Source: Author's computation.

Table 6 displays the log-log model S-GMM and QR analysis. Column 2 shows the system GMM regression result. The three-phase quantiles of regression, such as Q25, Q50, and Q75, are shown in columns 3 to 5. The coefficients of lnTO to explain lnFE in the QR models for Q25, Q50, and Q75 are 0.203, 0.385, and 0.407, respectively, and this estimation is positive and significant. Women's job opportunities grow by 0.203%, 0.385%, and 0.407% for every 1% increase in trade openness. Similarly, except for column 2, the coefficients of lnGDP to explain lnFE are positive. The coefficients are −0.0191, 0.232, and 0.367. Positive coefficients have a substantial impact. The negative coefficients of the lnURBA variable are −0.974, −1.192, and −1.819, showing that a 1% rise in urbanization provides a barrier to women's employment of 0.974%, 1.192%, and 1.819%, respectively. With coefficient weights of 0.0620, 0.0389, and 0.100, lnFDI has a favorable and significant influence on lnFE in all quantiles. Simultaneously, the empirical estimates of Q25, Q50, and Q75 show that lnME has a mixed effect on women's employment. Moreover, the first column of Table 6 presents the findings of the system GMM and indicates that trade openness, GDP per capita, and urbanization have a significant influence on the female employment rate. According to the system GMM, the coefficient values of lnTO and lnGDP of 0.085 and 0.232 are significantly positive. Thus, a 1% rise in lnTO and lnGDP will increase lnFE by 0.085%

and 0.232%, accordingly. The findings also show that lnURBA has a highly significant negative association with lnFE; a 1% increase in lnURBA will lower the lnFE by 0.692%.

**Table 8.** Impact of trade and other attributes on female employment in the industries.

| Variables | S-GMM | Q25 | Q50 | Q75 |
|---|---|---|---|---|
| L.lnINDUS | 0.885 | | | |
| | (0.682) | | | |
| lnTO | 0.278 | 0.0669 | 0.278 | 0.0550 * |
| | (0.217) | (0.143) | (0.217) | (0.0931) |
| lnGDPpc | 0.560 *** | 0.191 ** | 0.560 *** | 0.0285 |
| | (0.133) | (0.0875) | (0.133) | (0.0570) |
| lnME | 0.530 | 1.648 | 0.530 | −5.375 *** |
| | (1.823) | (1.202) | (1.823) | (0.783) |
| lnURBA | −0.379 | 0.484 ** | −0.379 | 0.0443 |
| | (0.300) | (0.197) | (0.300) | (0.129) |
| lnFDI | 0.0546 | 0.132*** | 0.0546 | −0.0124 |
| | (0.0546) | (0.0360) | (0.0546) | (0.0235) |
| Constant | −2.062 | 4.370 | −2.062 | 26.11 *** |
| | (8.153) | (5.375) | (8.153) | (3.500) |
| Observations | 158 | 173 | 173 | 173 |

Standard errors enclosed by brackets. *p*-value less than 0.1, 0.05, and 0.01 indicated by asterisk signs *, **, ***.
Source: Author's computation.

Quantile regression coefficients and confidence intervals for the relationships between FE, TO, GDP, URBA, FDI, and ME, and their interactions, are shown in Figure 5. Detailed results for a range of quantiles from the 0.00th to the 1.00th, with a 0.20-step interval, are provided. After appropriate smoothing, data are presented with estimated coefficients and a 95% confidence interval.

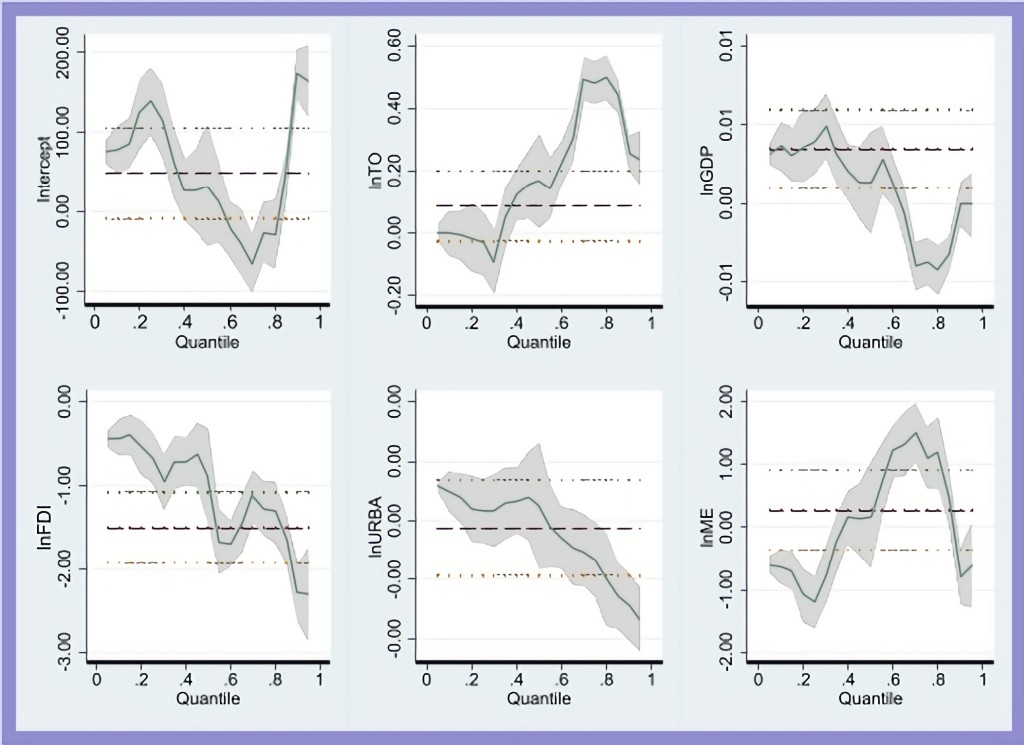

**Figure 5.** Quantile regression coefficients and confidence intervals in different quantiles. Source: Author's calculation.

Women's employment rates in various sectors are influenced by trade openness in the following ways:

$$FE_{j,i,t} = \alpha + \beta TO_{j,i,t} + \gamma X_{j,i,t} + \mu_i + \pi_j + \delta_t + \varepsilon_{i,t} \tag{8}$$

The above Equation (8) was used to assess the influence of trade liberalization on women's involvement in agriculture, service, and industry. The study incorporated the three sectors denoted by j, the countries represented by i, and the time frame indicated by t to produce a more meaningful equation.

Table 7 shows the effects of trade openness and other attributes on female participation in agriculture. Columns 3 to 5 display the different regression quantiles, including the three-phase quantiles, such as Q25, Q50, and Q75. On the other hand, column 2 shows the system GMM results. In the QR models, the three quantiles (Q25, Q50, and Q75) and the coefficients of lnTO explaining lnAGRI are −0.179 (for Q25), −0.299 (for Q50), and −0.142 (for Q75), respectively. These coefficients are negative and statistically significant. According to the study, women move from agricultural sectors by −0.179%, −0.299%, and −0.142% for every 1% increase in trade openness. Similarly, the S-GMM results show a 1% increase in trade when women move from agriculture to other sectors by −0.299%. Similarly, the coefficients of lnGDP to explain lnAGRI are all negative and significant. These values are represented by the coefficients −0.465, −0.665, −0.465, −0.116, and 0.0212. As a result, the higher the increase in GDP per capita, the greater the willingness of females to leave the agricultural sectors. In the lnURBA variable, the negative coefficients are −0.0347, −0.206, −0.0347, −0.0426, and −0.0968. The last quantile is the only one, which is statistically significant. All coefficients suggest that the rise of urbanization presents a barrier to women's employment in agricultural sectors. lnFDI positively impacts lnFE in all quantiles, with coefficient weights of 0.117, 0.0560, and 0.0153, and most of the quantiles are significant. On the other hand, the log of male employment negatively influences the first three quantiles and has favorable effects on the rest of the quantiles. In addition, the first column of Table 7 shows the findings of the system GMM and indicates that trade openness, GDP per capita, male labor force, and FDI have a significant influence on the female labor participation rate in agriculture. Table 7 demonstrates that lnTO, lnGDP, and lnME significantly negatively impact lnFE in agriculture. The result implied that a 1% rise in lnTO, lnLGDP, and lnME would lower the lnFE by 0.299%, 0.465%, and 3.360%, accordingly. The findings of the GMM also showed that a 1% increase in FDI will increase female employment by 0.0560%.

Table 8 demonstrates the outcomes of the log-log model QR analysis. The five-phase quantiles of regression, such as Q25, Q50, and Q75, are shown in columns 1 to 5. The coefficients of lnTO to explain lnINDUS in the QR models for Q25, Q50, and Q75 are 0.0669, 0.278, and 0.0550, respectively; this estimation is optimistic, and the upper quantiles are significant. On the other hand, the coefficients of lnGDP to explain lnINDUS are positive, and most of the quantiles are significant. The coefficients are 0.191, 0.560, and 0.0285, showing that a 1% rise in GDP per capita boosts women's employment in the industrial sector by 0.191%, 0.560%, and 0.0285%, respectively. lnURBA, lnFDI, and lnME have mixed impacts on lnINDUS in different quantiles. With coefficient weights of 0.132, 0.0546, and −0.0124, lnFDI positively impacts lnINDUS in the first three quantiles and negatively impacts the rest. Simultaneously, the empirical estimates of Q25 and Q50 show that lnME has a considerable impact on increasing women's employment, although Q75 shows a negative and significant impact on lnINDUS. In addition, the findings of system GMM are reported in the first column of Table 8. The results showed that GDP per capita significantly and positively impacted female labor involvement in the industry. The findings of system GMM showed that a 1% increase in lnGDP will increase lnFE in the industry by 0.560%.

The log-log quantile regression analysis is shown in Table 9 for the service sector. The three-phase quantiles of regression, such as Q25, Q50, and Q75, are shown in columns 3 to 5. The coefficients of lnTO to explain lnSERV in the QR models for Q25, Q50, and Q75 are 0.211, 0.204, and 0.285, respectively; this estimation is positive, and all quantiles are significant. Women's job opportunities in service sectors grow by 0.211%, 0.204%,

and 0.285% for every 1% increase in trade openness. Similarly, the lnGDP coefficients that explain lnSERV are positive and statistically significant. The coefficients are 0.292, 0.448, and 0.271, showing that a 1% rise in GDP per capita results in increases in female employment in the service sectors of 0.292%, 0.448%, and 0.271%, respectively. Except for the second quantile, the coefficients of lnURBA positively impact lnSERV. lnFDI has a negative and significant impact on lnSERV in all quantiles. Simultaneously, the empirical estimates of Q75 and Q95 show that lnME is a considerable obstacle to decreasing women's employment, although Q5, Q25, and Q50 positively impact lnSERV. The findings of system GMM are reported in the second column of Table 9. The result showed that lnTO and lnGDP positively impact female employment in service sectors, while lnFDI has a negative association with lnFE. The results also showed that a 1% increase in lnTO and lnGDP will increase lnFE in the service sector by 0.204% and 0.448%. The outcomes also showed that a 1% increase in lnFDI will lower the lnFE by 0.0663%.

**Table 9.** Impact of trade and other attributes on female employment in the service sectors.

| Variables | S-GMM | Q25 | Q50 | Q75 |
|---|---|---|---|---|
| L.lnSERV | 0.890 | | | |
| | (0.842) | | | |
| lnTO | 0.204 ** | 0.211 ** | 0.204 *** | 0.285 *** |
| | (0.0928) | (0.103) | (0.0928) | (0.0836) |
| lnGDPpc | 0.448 *** | 0.292 *** | 0.448 *** | 0.271 *** |
| | (0.0568) | (0.0631) | (0.0568) | (0.0512) |
| lnME | 0.472 | 2.468 *** | 0.472 | −0.473 |
| | (0.781) | (0.867) | (0.781) | (0.703) |
| lnURBA | 0.125 | −0.0247 | 0.125 | 0.383 *** |
| | (0.128) | (0.142) | (0.128) | (0.116) |
| lnFDI | −0.0663 *** | −0.0272 | −0.0663 *** | −0.104 *** |
| | (0.0234) | (0.0260) | (0.0234) | (0.0211) |
| Constant | −0.819 | 12.95 *** | −0.819 | 3.722 |
| | (3.491) | (3.875) | (3.491) | (3.144) |
| Observations | 159 | 173 | 173 | 173 |

Standard errors enclosed by brackets. *p*-value less than 0.05, and 0.01 indicated by asterisk signs ** and ***. Source: Author's computation.

## 5. Discussion

Figure 6 depicts how trade openness boosts the service and manufacturing sectors while shrinking the agricultural sector. Women are increasingly moving into service and manufacturing jobs as trade opens up new opportunities in the SAARC region. It is clear that trade has a positive and statistically significant effect on women's employment in service and industry but a negative effect on women's employment in the agriculture sector. It is well known that trade is fundamentally related to service- and industry-related sectors, so this encourages women to enter the service and industrial sectors. On the other hand, because the trade, tourism, and service industries in SAARC nations are developing, this is one of the primary reasons for this. When nations deal with one another, this expands the consumer demand for their goods and services. As a result, there may be an increase in the demand for goods and services, which in turn may lead to the creation of new employment opportunities for women [74]. Table 6 shows that increased GDP per capita and FDI have led to a rise in total female employment in SAARC nations. As GDP and FDI grow, the economy will expand, and more job opportunities will be open for women. Because of the increasing GDP, consumption is rising, production is climbing, new industries are opening, and tourism is also growing. All of these factors push women's overall employment. The results also reveal that urbanization is a barrier to women's employment in SAARC countries. In urban areas, women are reluctant to work, and urban women are more solvent than rural ones. Their labor forces are increasingly migrating to traditional agriculture and manufacturing. In addition, FDI is expanding, and ready-made garments and export-oriented businesses in SAARC are also increasing because of low wages. According to

Tables 6–8, FDI has a beneficial influence on female employment. The present research applied the impact of total FDI on disaggregated female employment in the labor market. The effects of FDI on juvenile labor were studied by Doytch et al. [75] who used both aggregate and sectoral data. In this study, we take into account the entire world, which includes all of the various continents. Based on the data, it appears that the economic effects of child labor vary across sectors. While foreign direct investment (FDI) in agriculture in Europe and central Asia tends to exacerbate child labor, FDI in manufacturing in south and east Asia, and FDI in mining in Latin America, appears to be inversely related to child labor. However, the current study examined the effect of total FDI on gender-segregated employment data in the labor market. Foreign direct investment has a beneficial effect on the number of women working in agriculture and industry sectors. In terms of women working in the service industry, FDI tends to have a detrimental effect. On the other hand, the agriculture sector's importance is decreasing daily in SAARC. Industry and service sectors pay more than the agriculture sectors. Additionally, agriculture is an informal sector, and there is no formal payment system in SAARC. On the other hand, industries and service sectors offer minimum wage and other benefits. As a result, a sizable workforce is moving from agriculture to industries and service sectors in SAARC. Therefore, overall, women are moving from agriculture sectors to industries and service sectors.

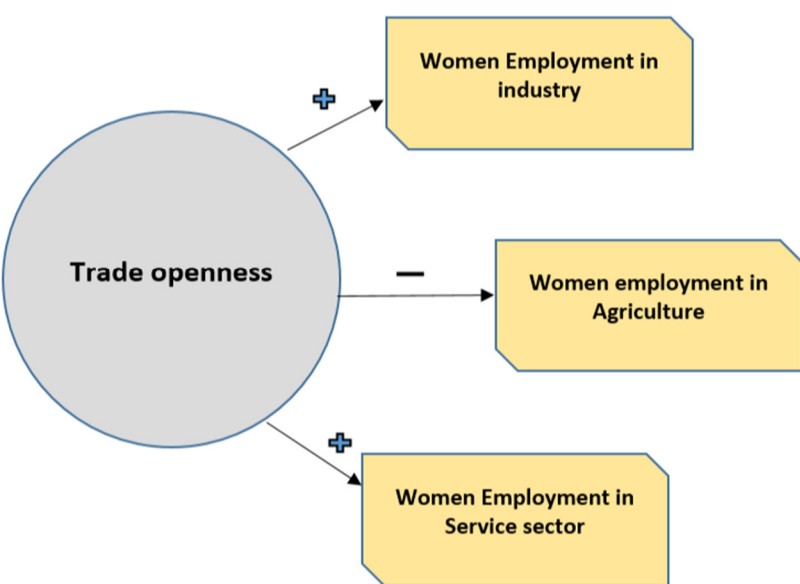

**Figure 6.** Effect of trade openness on female employment in three sectors. Source: Author's calculation.

## 6. Conclusions

This paper's main objective was to assess whether or not trade openness impacts female labor force participation rates in SAARC. The findings show that more trade openness resulted in a faster increase in the proportion of women in the labor force. The research also considered male employment, GDP per capita, FDI, and urbanization variables. These variables are highly linked with women's employment in SAARC. On the other hand, there are three sectors that women are working in: agriculture, industry, and service. The secondary objective of the research was to determine how trade openness impacts women's participation in these three sectors, as it is essential to find out which sector is now attracting women more. From agriculture to industrial and service industries, the female labor force is increasingly dominant in two sectors. Our research indicates that trade liberalization negatively affects women's labor force participation in agriculture but positively affects the services and industry sectors. Since the 1990s, SAARC countries have transformed into largely export-oriented economies, with trade openness expanding yearly. International trade is intimately tied to the service and manufacturing industries. As a result, trade liberalization propelled the industry and the service sectors. Regarding

SAARC, the proportion of agricultural output in GDP has declined, while the proportion of service and manufacturing output has climbed. Female laborers are transitioning away from agriculture and into these two industries simultaneously. According to the study, women move from agricultural sectors by $-0.179\%$, $-0.299\%$, and $-0.142\%$ for every 1% increase in trade openness. This study adds to the economic, policy, and scholarly conversation on the distribution, female empowerment, and gender consequences of trade openness. Regarding women's engagement in economic activity, trade openness is universally regarded as a positive indicator of progress. Similarly, increasing GDP per capita and FDI increase women's employment in SAARC countries. This means the more GDP and FDI are increasing, the more employment options for women are opening. The findings also show that urbanization hinders women's employment in SAARC areas. The empirical findings of this study demonstrate that trade liberalization has provided women with new opportunities. Governments and policymakers should open up new industrial and service sector frontiers, as women's labor forces have migrated to these sectors over the past few years. More research is needed to determine the influence of foreign direct investment and international business on the lives of women around the world.

## 7. Policy Recommendation

Based on the findings of this study, a few policy recommendations are suggested. Most of the obstacles to increasing female employment can be overcome through policy support from the government. The critical challenge for SAARC countries is efficient government policy formulation. Proper policy support can help the growth of job opportunities, creating more employment for women. Moreover, a decent work environment is essential for the workforce. The present export-oriented industries are attracting more women to the industrial sectors. However, there is a debate regarding the present export-oriented industry not providing proper work environment for their employees. Thus, policymakers and employee management should emphasize creating good workplaces for all employees, especially women. This policy will be beneficial for women's employment.

Increasing women's freedom and choice begins with better education and skill upgrades across the field. Preventing dropouts is a part of this. The next step is to make it illegal for employers to discriminate against women and to support policies that do so actively. Then, the workplace infrastructure must be improved to deal with restrictions and employee turnover. This includes aspects such as clean restrooms, a safe workplace, and following the rules for providing maternity leave and child care services. Finally, we must change our mindsets and modify our beliefs regarding female workers to eliminate discriminatory gender stereotypes. These actions will support greater female involvement in the labor force. The findings may achieve improved female labor force involvement and other objectives for women's empowerment through promoting women's financial inclusion, which is an important policy instrument. Microcredit, savings items, bank account provision, direct deposit salaries, psychometric grading in substitution of collateral, digital financial goods and services, and self-help groups are only some of the policy alternatives, which attempt to close the gender gap in women's access to finance. Some policy alternatives, such as psychological scoring, are new, while others, such as microcredit, have been the subject of extensive study. Microfinance, microcredit, and self-help groups have all been found to increase female engagement in the labor force. Therefore, more research must advise policymakers on the connections between financial inclusion and women's labor force participation. Some particular techniques, such as psychometric grading as collateral, need more excellent investigation. On the other hand, FDI creates more employment opportunities. SAARC countries should be prepared to attract more FDI. Attracting FDI requires governments to take their start-up ecosystems and value propositions seriously. FDI flows in from around the world when a country demonstrates that it has big consumer markets, free markets, stable governments, free trade zones, and high-quality infrastructure. Sometimes, SAARC countries face political instability. It is essential to affirm political stability to attract FDI. Utilizing capital and social networks may

help close market information gaps. Regarding the acquisition of social capital in general, the findings do not show that women are in a worse situation than men. Nevertheless, not all social relationships are equivalent. On the one hand, familial ties afford women the opportunity to join the labor force by providing childcare choices. On the other hand, weaker but more extensive relationships increase females' chances of joining the job market. This is particularly true when relationships are formed with influential people. Therefore, social capital policies should consider the variety and dispersion of these networks.

## 8. Limitations and Future Research

In this aspect, this study will contribute to future research in many ways. Firstly, the present structure of the research will be helpful for future researchers when designing trade and women's studies. In this research, missing data are the problem; future research will overcome this problem. Secondly, this study considers eight SAARC countries in evaluating the impact of trade, FDI, urbanization, male employment, and income on women's employment, giving us specific area perspectives. However, future research may investigate a single country to better understand the relationship between trade and women's employment. Lastly, future research can apply the present framework used in this study to conduct similar types of study in various regions. The current research applied GMM and QR for analysis; future research will apply more advanced models in order to obtain more accurate results. The effect of sectoral foreign direct investment (FDI) and sectoral trade on female employment in various sectors will be studied in the future by researchers.

**Author Contributions:** Conceptualization, E.N. and M.P.; Funding acquisition, S.B.; Methodology, T.K.; Supervision, E.N. and M.P.; Visualization, C.M.; Writing—original draft, C.A. and S.B.; Writing—review and editing, T.K., C.A. and C.M. All authors have read and agreed to the published version of the manuscript.

**Funding:** The research did not receive any funding.

**Data Availability Statement:** On-demand data availability.

**Acknowledgments:** To the esteemed editor and reviewer, the writers extend their gratitude.

**Conflicts of Interest:** The authors declare no conflict of interest.

## Appendix A

**List of Countries:** India, Afghanistan, Bangladesh, the Maldives, Nepal, Bhutan, Pakistan, and Sri Lanka are all SAARC members.

## Appendix B

| Abbreviation | Definition |
| --- | --- |
| SAARC | Group of Seven |
| GMM | South Asian Association for Regional Cooperation |
| S-GMM | System GMM |
| QR | Quantile Regression |
| GDP | Gross Domestic Product |
| FDI | Foreign Direct Investment |

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
