# Peer review of "Impact of Trade, FDI, and Urbanization on Female Employment System in SAARC: GMM and Quantile Regression Approach"

_systems, doi:10.3390/systems11030137_

Round 1

Reviewer 1 Report

The attached manuscript entitled "Impact of Trade, FDI, and Urbanization on Female Employment system in SAARC: GMM and Quantile Regression Approach" can be thematically characterized as attractive in terms of fulfilling the meaning of social demand. I definitely recommend it for publication after minor revisions.:

Abstract – It is written very matter-of-factly. The background of the research and the overall objective are briefly stated. Applied statistical methods and their results are also presented. The contribution of the results was not forgotten either.

Introduction – The text succinctly introduces the research issue of women's employment in SAARC countries. The sufficiency of sources in the text can be acknowledged. On the other hand, it would be good to divide it more into paragraphs for clarity. It is also advisable to "concentrate" the main goal and sub-goal more (i.e. the redundancy of the last sentence).

L. Review – Similar to the introduction, it is appropriate to divide the text into paragraphs for clarity. However, there could be more sources, despite the quality methodical guidance of the text. In this regard, we can mention (based on similar territorial aspect of research) the study by Vochozka et al. (2022):

Vochozka, Marek, Filip Petrach, and Svatopluk Janek. 2022. "Changes in Perception of Coffee in Eu: Luxury Good Turned Inferior". Economics & Sociology, 15(3), 248–67. doi: 10.14254/2071-789X.2022/15-3/14.

Application parts – Data and methods are very clearly explained. The application part itself confronts the methodology with key sources and findings in the first phase, which is beneficial. The statistical apparatus used (GMM and QR) can be described as sufficient to obtain quality results.

Discussion, Conslusion, Policy recommendation and Limitations – The discussion factually summarizes the results. For a better summary, it is advisable to mention only the most important sources and confront them with the established facts. The other chapters describe the conclusions of the research problem in detail and can be characterized as methodologically and substantively correct and concise.

In conclusion, the content of the manuscript can be evaluated as very interesting and beneficial. The defined socio-geographical area has been dealing with the issue of women's employment for a very long time. There have been many studies on this topic, and this manuscript will clearly contribute to the research material.

Author Response

Thanks, professor, for your valuable comments to improve our manuscript.  We are really grateful to you. The response was uploaded here.

Reviewer 2 Report

Impact of Trade, FDI, and Urbanization on Female Employment 2 System in SAARC: GMM and Quantile Regression Approach

Review

The paper examines an under-researched question: the impact of trade liberalization, urbanization, and foreign direct investment (FDI) on female participation in the labor force. The study is done on eight SAARC countries for the period 1991 to 2021 and differentiates between female labor participation in three distinct sectors- agriculture, industry, and services. The authors use first panel quantile regression (QR) and then- dynamic panel GMM. They find that total female employment benefits from increases in GDP and FDI, but not from urbanization. Further, women's participation in the services and manufacturing sectors increases, whereas their participation in agriculture decreases, as a result of increased trade openness.

The study is interesting and fills up a gap in the literature due to the fact that data on female labor participation, especially by sectors, is not easy to compile. The data in the current study is sourced from ILO.

Comments

1)      My main concern is related to the size of the data set, which has implications for the quality of the estimates. Table 2 lists the number of observations of the total female labor participation, but not the participation by sectors. Could the authors add this information?

2)      Further, since the dimensions of the panel data set are rather small, and especially the number of countries relative to the number of years is small, the instruments created by system GMM suffer certain weakness (Roodman, 2009). Could the authors address this question.

Roodman, D. (2009). How to do xtabond2: An introduction to difference and system GMM in Stata. The stata journal9(1), 86-136.

3)      May the authors be advised that in all of their equations, for example, eq. (1) and (2) the sign used for function appears in the printout as a sign for integral instead.

4)      When the “j” subscript is introduced in eq. (7), it should be clearly defined.

5)      It would be interesting for the sectoral regressions to be re-run with sectoral FDI and sectoral trade, if possible. Is data on sectoral FDI and trade available for these countries?

6)      As I pointed out, there are very few studies on this subject, due to lack of availability of employment data. However, there happens to be a study closely related the examined issue, which looks at the impact of sectoral level FDI on child labor (Doytch et al., 2014). The status of women and children regarding joining the labor force in economic downturns in developing countries is similar. The authors should compare and contrast their results with this study.

Doytch, N., Thelen, N. & Mendoza. R.U. (2014). The impact of FDI on child labor: Insights from an empirical analysis of sectoral FDI data and case studies. Children and Youth Services Review, 47(2), 157-167, 10.1016/j.childyouth.2014.09.008

7)      The authors should include time-series graphs on FDI vs. female employment and urbanization vs. female employment in addition to the presented graphs.

Author Response

Thanks, professor, for your valuable comments to improve our manuscript.  We are really grateful to you. The response paper was uploaded.

Round 2

Reviewer 2 Report

The authors have addressed my questions.